# Latent Space Simulator for Unveiling Molecular Free Energy Landscapes and Predicting Transition Dynamics

## Abstract

Free Energy Surfaces (FES) and metastable transition rates are key elements in understanding the behaviour of molecules within a system. However, the typical approaches require computing force-fields across billions of time-steps in a molecular dynamics (MD) simulation, which is often considered intractable when dealing with large systems or databases. In this work we propose LaMoDy, a latent-space MD simulator to effectively tackle the intractability with around 20-fold speed improvements compared to classical MD's. The model leverages a chirality aware $SE(3)$-invariant encoder-decoder architecture to generate a latent space, coupled with a recurrent neural network to run the time-wise dynamics. We show that LaMoDy effectively recovers realistic trajectories and FES more accurately and faster than existing methods, while capturing their major dynamical and conformational properties. Furthermore, the proposed approach can generalize to molecules outside the training distribution.

## 1 Introduction

Fundamental quantities of interest towards understanding a molecule's dynamics and properties are its Free Energy Surface (FES) and metastable states, alongside its transition rates between metastable states. Accessing them enables many real-world applications in drug discovery or material sciences (Peng et al., 2014; Bochevarov et al., 2013). Each 3D conformation of a molecule is associated with a potential energy that determines its probability of occurring (via a Boltzmann distribution).

The FES is a lower-dimensional representation of this energy landscape, providing insights into stable states (energy minima), transition pathways, and free energy differences. Additionally, a molecule's kinetics are of interest, such as the transition rates between metastable states/modes of the Boltzmann distribution.

The usual approach to compute these properties is to run long micro-second molecular dynamics (MD) simulations. Considering that each MD step is in the scale of femto-seconds, the simulation comes with a high computational cost. To accelerate the recovery of these properties, it is essential to develop a method that (1) can operate at time steps beyond the femtosecond level; (2) captures the key reaction coordinates; (3) does not suffer from instabilities (unphysical states) for long-time simulations.

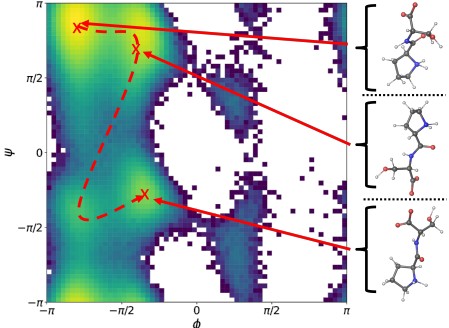

Figure 1: Free Energy Surface (FES) with minima corresponding to different conformations and an example MD trajectory as dotted arrow.

Learned simulators operating in a latent space suit these requirements if the latent space captures reaction coordinates (a molecule's most important degrees of freedom) since they allow for larger time steps (Sidky et al., 2020; Vlachas et al., 2022). However, existing architectures restrict the simulator to only work on a single molecule at a time, meaning that they cannot generalize to new molecules (Sidky et al., 2020; Vlachas et al., 2022).

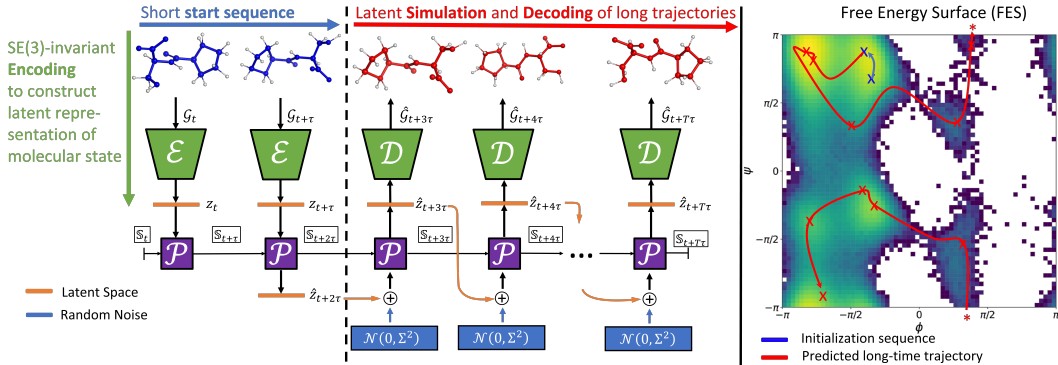

Figure 2: Overview of LAMODY. An encoder $\mathcal{E}$ computes $SE(3)$-invariant latent embeddings of a short initialization sequence, the dynamical propagator $\mathcal{P}$ iteratively predicts the next states to produce a long-time trajectory in latent space from which molecular conformers can be reconstructed by the decoder $\mathcal{D}$. The warm-up sequence and predicted trajectory are visualized in the FES. Here, $\mathcal{N}(0, \Sigma)$ denotes random noise, $\oplus$ is vector addition, $\mathcal{G}_t$ denotes the 3D graph representation of a molecule at time $t$, $z$ is a latent space state, $\tau$ is the time lag between states in a trajectory, and * denotes the point where the MD trajectory crosses the plane.

Furthermore, LED (Vlachas et al., 2022) fails to recover rare metastable states and lacks practical relevance as it has only been shown to work with multiple re-initializations from Boltzmann distributed states, meaning that a long MD simulation is still required to define the starting states.

Other approaches, such as Boltzmann generators (Noé et al., 2019) or Distributional Graphormer (Zheng et al., 2023) can predict the equilibrium distribution of unseen molecules but do not have a notion of time, i.e., no dynamical properties such as the transition rates can be extracted. In this regard, machine learning (ML) force fields (Unke et al., 2021; Batzner et al., 2022; Hu et al., 2021) have made significant progress for ab-initio simulations but are still slower for long simulations and larger molecules where classical force fields are applied (Fu et al., 2023).

To tackle these limitations, we propose a learned Latent Molecular Dynamics LAMODY, model. We employ an $SE(3)$-invariant encoder-propagator-decoder scheme based on message-passing neural networks (MPNN) (Gilmer et al., 2017) that can be trained end-to-end on MD data and can generalize to unseen molecules. For the tasks of FES recovery, past studies used different sampling and evaluation protocols, making it difficult to compare methods. We define scientifically meaningful tasks and metrics that allow that reflect a model's practical relevance in probing the free energy surface of molecules. In summary, our contributions are:

- 20-fold speed improvements compared to classical MD, thanks to a long operating time step of $100 fs$.
- Generalization to unseen molecules thanks to our chirality-aware $SE(3)$-invariant encoder-decoder.
- Defining a systematic evaluation scheme to assess the performance of simulation methods against scientifically meaningful tasks for FES recovery.

## 2 RELATED WORK

**Coarse Graining (CG)** is a classical approach towards enabling longer time-steps and faster simulation by grouping multiple atoms into coarse-grained beads. A CG mapping can be constructed through different approaches such as heuristics (Kmiecik et al., 2016), graph clustering (Fu et al., 2022) or autoencoders (Wang & Gómez-Bombarelli, 2019). The major shortcoming of coarse-graining methods is that (1) their effectiveness depends heavily on the system and observable of interest, and (2) they lose atomic details of the molecular system. Their accuracy is usually not sufficient for recovering the FES of flexible molecules. LAMODY generalizes across systems and provides all-atom structures.

**Enhanced sampling** methods inject bias to the potential energy function to facilitate fast sampling of transitions between local energy minima that are separated by high energy barriers. Popular methods include simulated annealing (Bernardi et al., 2015; Tsallis & Stariolo, 1996), metadynamics (Laio & Gervasio, 2008), replica exchange (Bernardi et al., 2015), umbrella sampling (Torrie & Valleau, 1977), and parallel tempering Yang et al. (2019). A major limitation of enhanced sampling methods lies in the fact that they typically require determining collective variables (CVs) in advance, which can be challenging for complex systems Wang et al. (2021). Furthermore, enhanced sampling methods do not have an explicit notion of "time", meaning that no extraction of dynamical properties is possible (Stelzl & Hummer, 2017).

**Latent Space Simulators** enable to accelerate MD simulations in the 3D configuration space, by updating a latent state generated by a learned encoder, instead of moving each atom according to its velocity and computed force. The updates are performed by a dynamical propagator, and the all-atom representation can be constructed with a decoder. Time-lagged autoencoders with propagators (Otto & Rowley, 2019; Lusch et al., 2018) learn a linear propagator whereas Sidky et al. (2020) use a mixture density network (Bishop, 1994) as a propagator. However, the above methods do not obey the $SE(3)$-invariance of molecules (they could, e.g., arbitrarily flip a chirality each step). Vlachas et al. (2022) train an LSTM network as propagator and account use a mixture density network as autoencoder. However, this method requires multiple re-initializations from Boltzmann distributed states and it remains unclear if the method stays stable for longer simulations. Additionally, all previously mentioned methods only work on a single molecule they have been trained on - they are not able to generalize unlike LaMoDy.

# 3 Method

Following the arguments of the previous section, we define an encoder-propagator-decoder framework that is generalizable across systems and obeys the $SE(3)$-invariance of molecular conformers.

To do so, we represent a molecule by its bond lengths $\mathbb{B}$, bond angles $\mathbb{A}$, and torsion angles $\mathbb{T}$ (internal coordinates), which provides an $SE(3)$-invariant representation of a molecular conformer. We employ an encoder to compute a fixed-sized latent embedding of the molecular state and use an LSTM (Hochreiter & Schmidhuber, 1997) to model the dynamics in the latent space. To regain molecular states with atomic details, we define a decoder that reconstructs the internal coordinates based on a latent state. We train the full model in an end-to-end fashion on MD data without restricting the latent space in any way, effectively allowing the model to construct meaningful latent representations capturing all important dynamical properties.

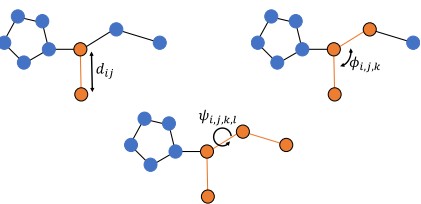

Figure 3: Internal coordinates of a molecule: Bond lengths $d_{ij}$; Bond angles $\phi_{i,j,k}$; Torsion angles $\psi_{i,j,k,l}$.

## 3.1 Model Architecture

**Encoder** To make the encoder architecture generalizable to other molecules, we use a graph representation of internal coordinates [1] and employ a Graph Neural Network (GNN) architecture. Concretely, a molecular state is represented by a graph $\mathcal{G} \in (\mathcal{V}, \mathcal{B}, \mathcal{X}, \mathcal{C})$ with each node representing a bond in the original molecule, and edges representing bond angles and torsion angles defined by triplets and quadruplets of bonds respectively, hence $|\mathcal{V}| = |\mathbb{B}|$ and $|\mathcal{B}| = |\mathbb{A}| + |\mathbb{T}|$. Nodes are featurized with information about the atoms forming the bond and the bond length and edges are featurized with the respective bond or torsion angle and a categorical feature indicating whether the edge defines a bond angle or a torsion angle [2]. We then employ $L$ message-passing layers akin to Shi et al. (2021), pool the nodes using a learnable set-to-set mapping (Vinyals et al., 2016), and predict the final latent vector using a linear layer.

---

[1]See Figure 3

[2]for a detailed description see subsection C.1

**Decoder** To reconstruct the internal coordinates of a molecular state given a latent representation, we use a second GNN similar to Winter et al. (2021). The decoder takes as input a two-dimensional molecular graph with nodes representing atoms and edges representing bonds and a latent vector describing the molecular state in the latent space. First node level embeddings are computed by iteratively applying a sequence of message-passing layers similar to the encoder. Then, bond lengths are predicted by applying a three-layer MLP onto the concatenated pairs of nodes and the latent embedding, i.e. $d_i = \Pi_{bond}([h_a, h_b, z])$ with $h_*$ being the node embeddings, $z$ the latent vector and $\Pi_{bond}$ the MLP. The same approach is taken for bond angles and torsion angles with triplets/quadruplets of node embeddings and $\Pi_{ang}.\Pi_{tor}$ respectively.

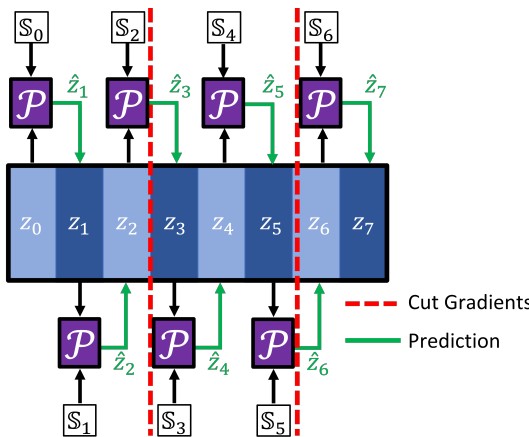

Figure 4: Training scheme for long sequences: The propagator $\mathcal{P}$ takes in a latent state $z_t$ and cell state $\mathbb{S}_t$ to predict the latent state at time $t+1$. The cell states are not re-initialized and gradients are detached after a fixed-length interval.

**Dynamical Propagator** As suggested by Vlachas et al. (2022), sequences of MD states are not necessarily Markovian since complex systems can exhibit long-term correlations in their behavior, meaning that future states can depend on past states, violating the assumption of independence between time steps. To account for this, we use an LSTM (Hochreiter & Schmidhuber, 1997) as the dynamical model that is trained to predict the next latent state given a short history. Concretely, we use

$$(h_{t+\tau}, c_{t+\tau}) = LSTM\,(z_t, h_t, c_t)$$
$$z_{t+\tau} = \Xi(h_{t+\tau}) \tag{1}$$

where $h_t, c_t$ denote the LSTM hidden state and cell state at time $t$, $z_t$ is the latent state at time $t$ and $\Xi$ is a two-layer MLP.

## 3.2 TRAINING

We train our model end-to-end on MD data. To do so, we randomly sample a batch of starting points from the dataset from which we consider the consecutive $k$ states with a time lag $\tau$ between states. Hence, we end up with a batch of sub-sequences of the full trajectory of length $k+1$ states. Starting with an initial LSTM state of $\mathbb{S}_0 = (h_0, c_0) = (\vec{0}, \vec{0})$, we iteratively unfold the LSTM to predict the next time step, while the LSTM cell states are passed through time. More specifically, we encode $\mathcal{G}_0$ into latent space by $z_0 = \mathcal{E}(\mathcal{G}_0)$, from which together with $\mathbb{S}_0$ the next time step latent state $\hat{z}_1$ is predicted. Then $\mathbb{S}_1$ and $z_1 = \mathcal{E}(\mathcal{G}_1)$ are used to predict $\hat{z}_2$, which can all be decoded back to molecular states.

To optimize the parameters of the model with backpropagation, we define an end-to-end propagation loss that is additionally regularized by a reconstruction loss and a latent loss :

$$\mathcal{L} = \delta_{e2e}\frac{1}{k}\sum_{i=1}^{k}\mathcal{L}_{rec}\left[\mathcal{G}_i, \mathcal{D} \circ \mathcal{P} \circ \mathcal{E}(\mathcal{G}_{i-1})\right]$$
$$+ \delta_{lat}\frac{1}{k}\sum_{i=1}^{k}||z_i - \hat{z}_i||^2 + \delta_{rec}\frac{1}{k+1}\sum_{i=0}^{k}\mathcal{L}_{rec}\left[\mathcal{G}_i, \mathcal{D} \circ \mathcal{E}(\mathcal{G}_i)\right] \tag{2}$$

here $\delta_{rec}, \delta_{lat}, \delta_{e2e}$ are hyperparameters and $\mathcal{L}_{rec}$ is defined as in Equation 11. Note that $z_i = \mathcal{E}(\mathcal{G}_i)$, $\hat{z}_i = \mathcal{P} \circ \mathcal{E}(\mathcal{G}_{i-1})$. Although the end-to-end part of our loss function theoretically encapsulates the latent and the reconstruction loss, we found the explicit presence of both as additional regularization to be crucial for the training process to succeed.

**Training on long sequences** As we aim to predict long-timescale trajectories at inference time with $N_{steps} \gg k$, we require training on long sequences without suffering from vanishing or exploding gradients. To do so, we sample sub-trajectories of length $c * k$ with $c$ being a hyperparameter and iteratively train on sequences of length $k$ where we keep the LSTM states but detach the gradients as suggested by Vlachas et al. (2022).

## 3.3 INFERENCE

At inference time, we "warm up" the LSTM with a sequence of $k$ MD states from which we iteratively unfold the propagator to predict latent trajectories. Additionally, we infuse artificial noise to the latent states before feeding them into the propagator. We found this to be crucial because otherwise, the dynamical model was prone to become stuck at a local energy minimum. Concretely, we predict the next latent state by :

$$\hat{z}_{t+\tau} = \begin{cases} \mathcal{P}\left(\hat{z}_t + \mathcal{N}(0, \Sigma)\right), & \text{if } x \sim U(0,1) \leq \beta \\ \mathcal{P}\left(\hat{z}_t\right), & \text{else} \end{cases} \tag{3}$$

where $\beta \in [0, 1]$ is a hyperparameter, $x \sim U(0, 1)$ indicates a sample from the uniform distribution and $\Sigma = \boldsymbol{I} * \sigma^2, \sigma^2 \in \mathbb{R}^+$ is computed from the warmup trajectory.

## 4 EVALUATION PROTOCOL FOR FES RECOVERY

This section aims to provide an evaluation protocol that is both robust and scalable. After identifying the issues with prior metrics, we propose a method of identifying metastable states and measuring the agreement between the model and the ground truth.

**Deficiencies of Past Metrics** Past studies have used different tasks and metrics for evaluation, making it difficult to compare methods. The metastable states of the free energy surface are frequently used for evaluation as they allow to reason about dominant conformations and transition rates. However, previous evaluation protocols are often not applicable to multiple systems but only allow qualitative inspection of single molecules at a time. To overcome these challenges, we propose a systematic evaluation protocol to reliably assess the quality of predicted trajectories for multiple systems.

A common practice to evaluate the quality of predicted FES is to use Kullback-Leibler (KL) divergences, either between one-dimensional marginals or the two-dimensional histogram (Klein et al., 2023). However, this method is heavily dependent on the chosen bin size of the histogram and ignores the fact that variations in the estimated density are negligible for multiple practical applications, where the correct identification of modes and transition rates is the desired goal.

Work on conformation generation (Jing et al., 2022; Zhu et al., 2023) is typically evaluated by computing the coverage of predicted structures (in terms of RMSD) and reporting precision and recall, i.e. the fraction of correctly predicted structures and the fraction of identified structures compared to MD. Similar to the KL-based metrics, this protocol does not capture whether modes and transition rates are correctly identified.

**Identifying metastable states** Identifying modes in a two-dimensional FES is highly non-trivial. While previous works used K-MEANS clustering to identify metastable states (Pandey et al., 2023; Jain & Stock, 2012), we found that K-MEANS frequently converges to incorrect minima. Therefore, we use the method of Novelli et al. (2022) where the FES is first smoothed using a Gaussian kernel and local minima are identified via running multiple BFGS solvers from random starting points. For a detailed explanation, we refer to subsection B.1. Lastly, the identification of reaction coordinates varies across past methods where multiple methods a sophisticated scheme such as Time-Independent-Component-Analysis (TICA) (Pérez-Hernández et al., 2013) to define the reaction coordinates from which the FES is constructed (Sidky et al., 2020; Klein et al., 2023). While TICA is useful for a variety of applications, it requires a Chapman–Kolmogorov test and manual inspection of the lag time to guarantee high-quality dimensionality reduction. Therefore, we use the two dihedral angles $\phi, \psi$ as they are known to capture the conformation space of peptides (Choudhuri, 2014).

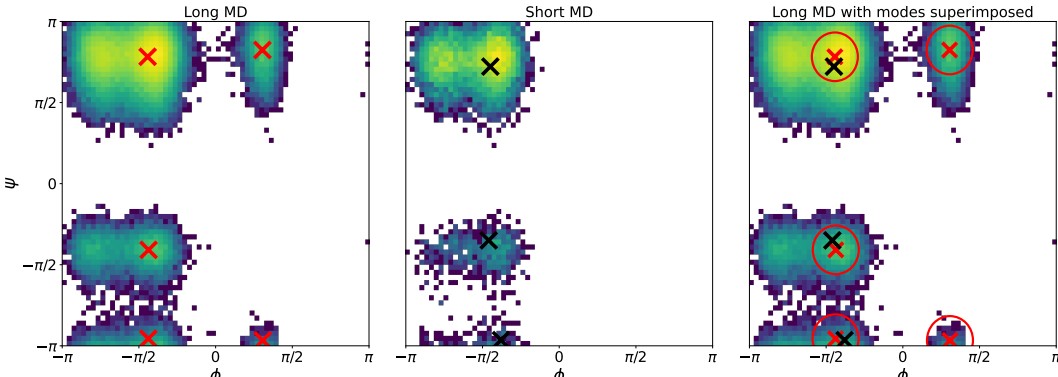

Figure 5: MSPR: Metastable State Precision/Recall; Ramachandran plot of a long and a short MD simulation for a peptide where identified metastable states are indicated by crosses. The third figure shows the long MD trajectory with modes identified by the short MD simulation superimposed and the circles denote the area where a mode is considered to be correct. This allows to compute metastable state precision and recall (MSPR).

**Metrics** With the above-described procedure, we can identify metastable states without the need of manual specification. This allows to compute precision and recall in terms of found metastable states, i.e. the fraction of correctly predicted modes and the percentage of modes found where a mode is considered correct if it lies within close proximity to the ground-truth MD mode [3].

Furthermore, the transition rates between these identified metastable states are relevant for many applications, such as inferring relaxation times or reaction rates, and can be studied using a Markov State Model (MSM) (Bowman et al., 2014). Hence, an MSM can be fitted to predicted and MD trajectories, allowing to compare transition rates. Specifically, the Mean First Passage Times (MFPTs) (Hoel et al., 1986) can be computed which represent the expected times for a transition to happen from a predefined origin state to a target state. The relative error across the MFPTs for multiple molecules compared to MD then gives insight about the practical use of the predicted dynamical properties.

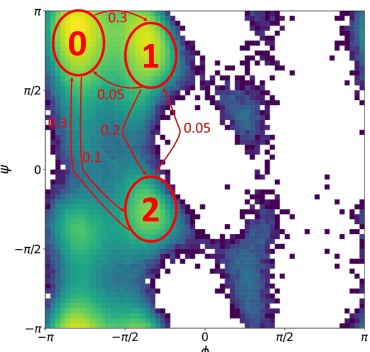

Figure 6: Example MSM with three states fitted to MD trajectory with transition probabilities.

## 5 EXPERIMENTAL RESULTS

In this section, we first show LAMODY's ability to recover the dynamics and transition states of alaninde dipeptide, then show that it effectively generalizes across peptides. We further demonstrate the large benefits of LAMODY in terms of simulation speed. Finally, we do ablation studies on some of the architectural choices.

### 5.1 ALANINE DIPEPTIDE

Before we evaluate the generalization capabilities to unseen molecules, we test our method on a single molecule, namely alanine dipeptide (ALDP), which is a widely used benchmark for MD simulators and has been the subject of evaluation in previous works. In the case of ALDP, the primary degrees of freedom under consideration are the two backbone dihedral angles $\phi$ and $\psi$. Despite the model being trained on this exact molecule, it's important to note that recovering long-time FES and transition rates remains highly nontrivial, as dynamical models are typically designed

---

[3]See subsection B.1 and Figure 5

to predict single or a limited number of steps. Specifically, we train on $100ns$ of MD data of ALDP in implicit solvent to assess whether the model can qualitatively reproduce the free energy surface in terms of the backbone dihedral angles. Additionally, we analyze the model's ability to predict transition rates between the identified metastable states, comparing them to MD results.

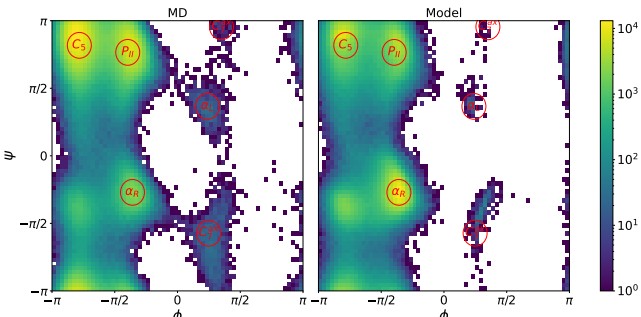

Figure 7: Ramachandran plots of trajectories from MD data and predictions of our model for alanine dipeptide with corresponding metastable states as defined by Vlachas et al. (2022).

**FES recovery** To use the trained model for simulating MD trajectories, we use the procedure described above. Starting from an initialization sequence of five states, we simulate a trajectory of length $100ns$ without re-initialization. The Ramachandran plots of the predicted trajectory alongside the MD simulation are visualized in Figure 7. Figure 7 shows that our model is able to capture all metastable states without becoming unstable, i.e. no unphysical states are visited throughout the entire simulation. Notably, the model is able to explore the rare states $C_7^{ax}, \alpha_L$, which previous latent space simulators (Vlachas et al., 2022) failed to achieve. The Ramachandran plots also show that our model slightly overestimates the density of $\alpha_R$.

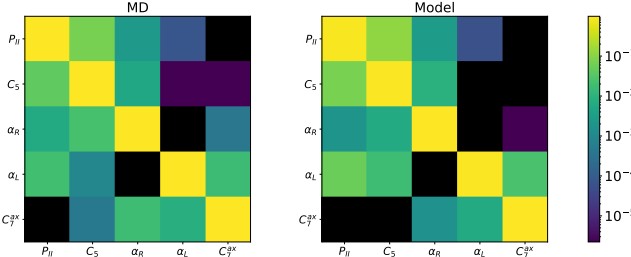

Figure 8: Transition probabilities of MSMs for alanine dipeptide estimated from MD data and predictions of our model. Black squares are transitions that were never observed.

**Transition dynamics** To examine whether the overestimation of $\alpha_R$ leads to unrealistic dynamical properties, we can compare the transition rates extracted from MSMs fitted to MD data as well as the predicted trajectory, which are shown in Figure 8. The transition probabilities clearly show that the dynamical properties that can be inferred from the model predictions closely match the true dynamics. Even for the highly unlikely states, our model approximates the correct transition rates. We found the training scheme for long trajectories as described above to be crucial for this.

## 5.2 GENERALIZATION ACROSS MOLECULES

After this first sanity check, we assess the capability of our approach to generalize to unseen molecules. To do so, we constructed a dataset of 216 dipeptides[4] with a length of $12ns$ each of which 200 are used for training and 16 are held out for evaluation. We use the systematic evaluation protocol introduced in section 4.

---

[4]Peptides with two amino acids

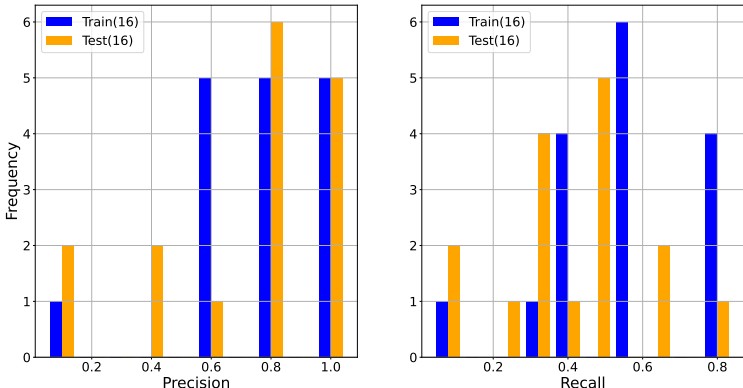

Figure 9: Metastable state precision and recall (MSPR) for train and test samples of the dipeptide model.

**FES recovery** In contrast to prior work on latent space simulators (Sidky et al., 2020; Vlachas et al., 2022) where the model can only be evaluated on the same molecule it has been trained on, our architecture is not restricted to single molecules. We evaluate the peptide model on 16 unseen molecules and randomly choose 16 peptides from the training set as a comparison. Figure 9 shows the precision and recall values the dipeptide model achieved. We can observe, that the model is better in terms of precision than recall. This suggests, that the learned simulator is more "conservative" and avoids predicting unphysical modes rather than exploring the full state space which is desirable. However, Figure 9 also shows that the model fails to recover the correct metastable states for a subset of the peptides.

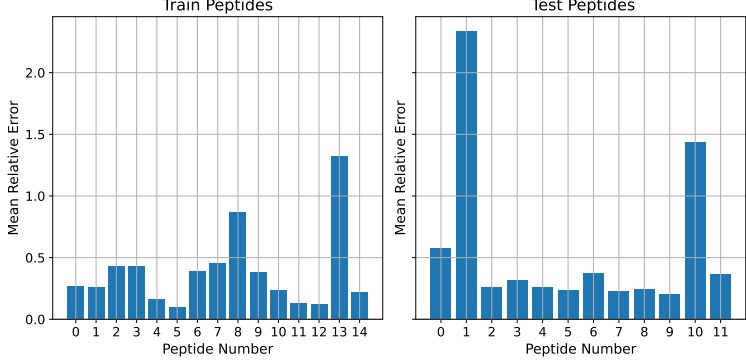

Figure 10: Mean relative error of MFPTs for MSMs fitted to predicted trajectories compared to MD for train and test set. Correctly extracted metastable states from the predicted trajectory are used to construct MSMs on MD and predicted data. Peptides where only one metastable state exists and therefore the MFPT error would always be zero are held out.

**Predicting transition dynamics** To gain more insight into the predicted trajectories, we evaluate the relative error between predicted and MD MFPTs for MSMs constructed from correctly identified states as defined in section 4. The results of this analysis are shown in Figure 10 where peptides that only contain one mode are excluded, as the MFPT error would be 0 in this case (only one state in the MSM, so no transitions). Figure 10 shows that the mean relative error is below 0.5 except for two peptides from the training set and two peptides from the test set. This confirms the previous results, i.e. that the model can approximate the majority of peptides very well, but misses a small subset. Furthermore, this metric shows that the modes which are found by the model are captured accurately and the transitions between the modes are captured within a relative error that existing latent space simulators (Vlachas et al., 2022) achieve for a single molecule they have been trained on. Furthermore, this shows the practical use of this method, as it can quickly and efficiently recover

the leading states of unseen molecules from which accurate transition rates can be extracted making this model especially useful for screening large chemical spaces.

## 5.3 SIMULATION SPEED

As high computational complexity/ slow simulation speed is the major limitation of MD simulations Table 1 shows the propagation speed of our method and MD in terms of iterations per second and the total wallclock time the respective simulation requires[5]. Table 1 clearly shows the advantage of our method that realizes a speedup of approximately 20, improving upon the results of Vlachas et al. (2022), who reported an acceleration by a factor of 3. Furthermore, in contrast to prior work, our model does not require re-initialization paired with short timescale predictions but can instead simulate long timescale trajectories starting from a five-state sequence without becoming unstable. Note that the predictions of our model can also be run in parallel with up to 128 peptides on a single GPU.

Table 1: Simulation Speed of MD and LAMODY given as averaged iterations per second and total wallclock times.

| Molecule | iteration/second | | wallclock time [minute] | |
|---|---|---|---|---|
| | MD | LAMODY | MD | LAMODY |
| ALDP | 189 | 3788 | 88 | 4.6 |
| Peptides | 117 | 2239 | 34.2 | 1.8 |

## 5.4 MODEL VARIATIONS AND ABLATIONS

**Cartesian Encoders** As the natural choice for an input representation seems to be representing a state by the two-dimensional molecular graph and associated cartesian positions, we also employed an $SE(3)$-invariant encoder operating on cartesian coordinates based on Euclidean graph neural networks (Geiger & Smidt, 2022). Additionally, we also used the popular GEMNET (Gasteiger et al., 2021) as our encoder network since GEMNET operates on cartesian coordinates and uses the internal coordinates of a molecule as features during message passing. However, we unexpectedly encountered that the cartesian encoder as well as GemNet failed to identify rare metastable states. The results of these simulations are shown in Figure 12 and Figure 13. We suspect this to be the case as both models are more memory intense than the internal encoder and we, therefore, had to reduce the length over which we unroll the propagator states during training [6].

## 6 DISCUSSION

We present MSPR, a reliable evaluation metric for FES that tackles the necessity of comparable evaluation schemes for learnerd simulators. Additionally, we introduce LAMODY, a learned simulator operating in a latent space to efficiently recover free energy surfaces and transition rates. LAMODY is trained end-to-end on MD data constructing its own latent space. The model employs an $SE(3)$-invariant encoder-propagator-decoder scheme. We show that our method can operate at integration time steps that are two orders of magnitude larger than for MD while still being able to conduct stable long-timescale simulations required for recovering properties such as FES and transition rates.

In contrast to prior works, LAMODY does not require re-initialization throughout the simulation, removing the need for prior MD simulations. We demonstrate that the predicted trajectories closely match the results of MD and correct dynamical properties can be recovered even for rare metastable states. Furthermore, our model is generalizable to molecules outside its training distribution and can capture their leading structural and dynamical properties. Overall, our approach is approximately 20 times faster at recovering FES and transition rates than classical MD and can additionally easily be parallelized for up to 128 peptides on a single GPU.

---

[5]Hardware specifications are reported in Appendix F

[6]Unrolling propagator states for long trajectories with detaching gradients, see subsection 3.2 for details.

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

## A    ADDITIONAL EXPLANATIONS

### A.1    MOLECULAR DYNAMICS SIMULATION

Molecular Dynamics (MD) simulations are a computational tool that can be utilized to study the behavior of molecules over time at an atomistic resolution. To do so, a popular method is Langevin Dynamics (Lemons & Gythiel, 1997), which evolves the positions and velocities of the system under study by the following stochastic differential equation:

$$m_i \frac{d^2 \boldsymbol{x}_i}{dt^2} = -\nabla_i U(\boldsymbol{x}_1, ..., \boldsymbol{x}_N) - \gamma m_i \frac{d\boldsymbol{x}_i}{dt} + \sqrt{2 m_i \gamma k_B T} dB_t \tag{4}$$

where $\boldsymbol{x}_i$ denotes the position of atom $i$, $U$ is the potential energy, $\gamma$ is a friction constant, $m_i$ is the mass of atom $i$, $T$ is the temperature of the system, $k_B$ is the Boltzmann constant, and $dB_t$ is standard Brownian motion. To ensure the stability of the simulation, the integration time step size is typically chosen to be in the range of a few femtoseconds. The potential energy of the molecule based on the coordinates of the particles $U(\boldsymbol{x}_1, ..., \boldsymbol{x}_N)$ is usually parameterized by a force field[7]. Machine learning methods that aim to simulate molecular systems are normally evaluated by their ability to recover conformational modes, free energy surfaces, and dynamical properties in comparison to a classical MD simulation (Vlachas et al., 2022; Sidky et al., 2020; Klein et al., 2023).

### A.2 INTERNAL COORDINATE GRAPH

Figure 11 shows the a visualization of the internal coordinate graph used by the encoder as defined in subsection 3.1.

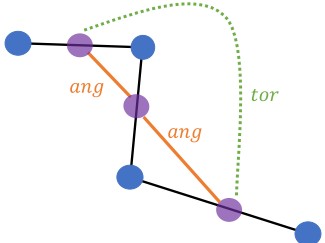

Figure 11: Graph of internal coordinates superimposed onto the molecular graph. Blue vertices and black edges show the corresponding molecular graph. The internal graph is superimposed with bond vertices in purple, bond angle edges in orange, and torsion angle edges in green.

## B ADDITIONAL RESULTS

### B.1 IDENTIFICATION OF METASTABLE STATES

Following Novelli et al. (2022), we use a standard Gaussian kernel density estimator (Scott, 1992) to approximate the free energy surface in the space of the two dihedral angles $\phi, \psi$ that are known to capture the conformational space for peptides (Choudhuri, 2014). Then we aim to identify the local minima of the FES as these will represent the metastable states. To do so, 100 BFGS solvers (Nocedal & Wright, 2006) are initialized at random points and run until convergence from which we recover the unique local minima. By doing so, we are able to reliably identify metastable states without the need for manual specification[8].

To assess the quality of our predictions, we apply this procedure to the trajectories produced by our model as well as the MD data. This allows to compute precision and recall of the metastable states extracted from the predicted trajectories where we consider a metastable state to be correctly identified if $||\mu_{pred} - \mu_{MD}|| \leq 0.15$. This allows us to judge the models' ability to recover correct FES for multiple peptides. Additionally, we use the set of correctly identified metastable states (from our model predictions) to construct an MSM for which we can compare the mean first passage times (MFPT) (Hoel et al., 1986) between MD and our model. The MFPTs are the expected time for a transition to happen from a predefined origin state to a target state. In practical applications this property is of great interest and can, for instance, be used to estimate the time it takes for a molecule to bind to a receptor. With this evaluation metric, we can judge the quality of the predicted dynamics and the practical use of the model, even if the model did not find all metastable states.

### B.2 MODEL VARIATIONS AND ABLATIONS

Figure 12 and Figure 13 show the inference results for the models with a cartesian/GEMNET encoder respectively. The figures show that both models miss the rare metastable states, which we suspect to

---

[7]see González (2011) for a detailed definition.

[8]An example is shown in Figure 5

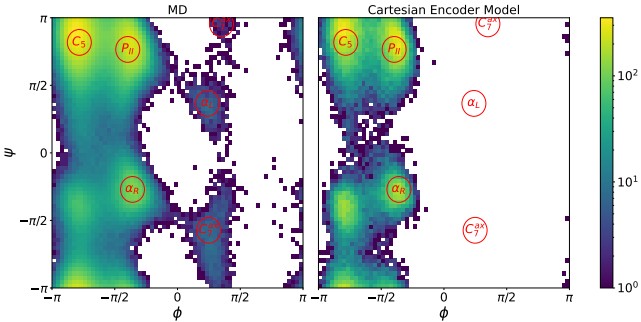

Figure 12: Ramachandran plots of trajectories from MD data and predictions of the model with cartesian encoder based on tensor product convolutions (Geiger & Smidt, 2022).

be caused by the shorter training sequences due to memory limitations as described in subsection 5.4.

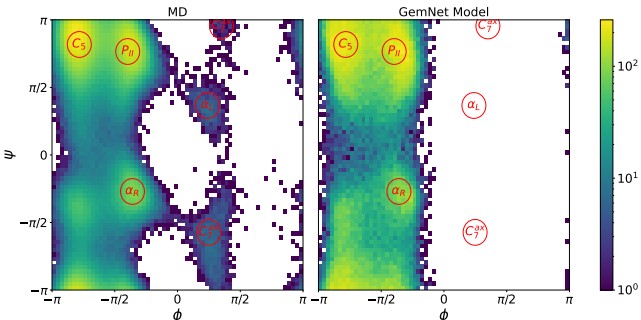

Figure 13: Ramachandran plots of trajectories from MD data and predictions of the model with GEMNET (Gasteiger et al., 2021) encoder.

## C  ARCHITECTURE DETAILS

### C.1  ENCODER

The internal encoder operates on the internal coordinate graph as described in subsection 3.1, which is $SE(3)$-invariant by construction. The internal coordinates are normalized to lie in $[0, 1]$.

Nodes $v_i$ are featurized with: Atomic number of the first atom in the bond, atomic number of the second atom in the bond, bond length, mass of the first atom, and mass of the second atom. Edges between all pairs of bonds that form a bond angle are featurized with the bond angle and an additional categorical feature indicating the edge type. Torsional edges are featurized with the torsion angle and the categorical feature accordingly. These scalar features are transformed by a set of learnable MLPs (one for each feature), to compute an initial feature embedding $h^0$ for each node.

After computing the initial embeddings $h_i^0$, we iteratively apply $L$ message passing layers that additionally employ a (multi-head) dot product attention mechanism to scale messages according to their importance, akin to Shi et al. (2021). More specifically, node embeddings for a node $a$ at layer $l$ get updated by:

$$h_a^{l+1} = \beta_a \boldsymbol{W}_1 h_a^l + (1 - \beta_a) \underbrace{\left( \sum_{b \in \mathcal{N}(a)} \alpha_{ab} \left( \boldsymbol{W}_2 h_b^l + \boldsymbol{W}_6 \boldsymbol{c}_{ab} \right) \right)}_{\boldsymbol{m}_a} \tag{5}$$

with

$$\alpha_{ab} = softmax\left(\frac{\left(\boldsymbol{W}_3\boldsymbol{h}_a^l\right)^T\left(\boldsymbol{W}_4\boldsymbol{h}_b^l + \boldsymbol{W}_6\boldsymbol{c}_{ab}\right)}{\sqrt{d}}\right) \tag{6}$$

$$\beta_a = sigmoid\left(\boldsymbol{W}_5\left[\boldsymbol{W}_1\boldsymbol{h}_a^l, \boldsymbol{m}_a, \boldsymbol{W}_1\boldsymbol{h}_a^l - \boldsymbol{m}_a\right]\right)$$

here $W_*$ indicates learnable parameters, $d$ is the hidden size of the attention heads, $[a, b]$ indicates vector concatenation, $\boldsymbol{c}_{ab} \in \mathcal{C}$ are the edge features of edge $(a, b)$, and $\mathcal{N}(a) = \{b|(a, b) \in \mathcal{B} \vee (b, a) \in \mathcal{B}\}$. Between each of the layers, ELU nonlinearities and batch normalization are applied. After the final message passing layer, we use a learnable set-to-set mapping Vinyals et al. (2016) to pool the nodes:

$$\boldsymbol{q}_t = LSTM(\boldsymbol{q}_{t-1}^*)$$
$$e_{i,t} = \boldsymbol{h}_i^L \cdot \boldsymbol{q}_t$$
$$\alpha_{i,t} = \frac{exp(e_{i,t})}{\sum\limits_j exp(e_{j,t})} \tag{7}$$
$$\boldsymbol{r}_t = \sum_{i=1}^N \alpha_{i,t}\boldsymbol{h}_i^L$$
$$\boldsymbol{q}_t^* = [\boldsymbol{q}_t, \boldsymbol{r}_t]$$

where $\cdot$ denotes the dot product and $\boldsymbol{h}_i^L$ indicates the node embedding after the final message passing interaction layer. This layer iteratively updates the aggregated set for $T$ processing steps by computing a weighted sum $r_t$ of node embeddings, concatenating this sum to the last state $\boldsymbol{q}_t$ and passing this concatenated vector $\boldsymbol{q}^*$ through the LSTM. We found this learnable set-to-set mapping to yield better results compared to sum or mean reduction. After the set-to-set aggregation, we use a linear layer $\Phi$ to map to the fixed-size latent embedding vector:

$$\boldsymbol{z} = \Phi\left(\boldsymbol{q}_T^*\right) \tag{8}$$

Given this model architecture, we are able to learn a mapping to a latent space, which is by construction of the graph $SE(3)$-invariant. Moreover, the model is not limited to a fixed-size graph but can be applied to graphs of distinct molecules.

## C.2 Decoder

The molecular decoder acts as a counterpart to the encoder and reconstructs a molecular state from a latent representation by predicting the molecule's internal coordinates for that state. The decoder architecture was heavily inspired by the work of Winter et al. (2021). As the decoder has to be applicable to different molecules, we condition the decoder on the time-invariant two-dimensional molecular graph. Concretely, the decoder predicts a molecular state at time t via:

$$\mathcal{G}_t = \mathcal{D}\left(\boldsymbol{z}_t, \mathcal{G}_{mol}\right) \tag{9}$$

To do so, we first compute node embeddings for all atoms of $\mathcal{G}_{mol} \in (\mathcal{V}_{mol}, \mathcal{B}_{mol}, \mathcal{X}_{mol}, \mathcal{C}_{mol})$ where nodes represent atoms and edges represent bonds between atoms in the molecule. $\mathcal{G}_{mol}$ is constant throughout and MD simulation, as only the atom position change. We featurize nodes with the following attributes: Atomic number, chirality, degree, number of rings the atom is involved in, implicit valence, formal charge, number of bonded hydrogens, hybridization type, whether or not it is in an aromatic ring, whether or not it is in a 5 or 6-ring, the residue name and the atom name. Bonds between atoms are featurized by bond type and a radial basis embedding of the bond length (Schütt et al., 2017). Since torsion angles are defined by quadruplets of atoms that do not necessarily have to be direct neighbors, we add additional edges by connecting each node to all its k-hop neighbors. Concretely, we modify $\mathcal{B}_{mol}$ to be $\mathcal{B}_{mol} := \{(a, b) \mid a \in \mathcal{V}_{mol} \wedge b \in \mathcal{N}^k(a)\}$ where $\mathcal{N}^k(a)$ denotes all nodes that can be reached with a maximum of $k$ hops from $a$. The additional edges facilitate the information flow over longer distances during message passing.

After an initial node embedding akin to subsection C.1, we apply $L$ message passing layers that update the node embeddings similar to subsection C.1. With the final node embeddings $\boldsymbol{h}_i^L$, we predict the internal coordinates of the current state by:

$$
\begin{aligned}
d_{ab}^t &= \Pi_{bond}\left(\left[\boldsymbol{h}_a^L, \boldsymbol{h}_b^L, \boldsymbol{z}_t\right]\right) \forall (a,b) \in \mathbb{B} \\
\phi_{abc}^t &= \Pi_{ang}\left(\left[\boldsymbol{h}_a^L, \boldsymbol{h}_b^L, \boldsymbol{h}_c^L, \boldsymbol{z}_t\right]\right) \forall (a,b,c) \in \mathbb{A} \\
cos\psi_{abcd}^t &= \Pi_{tor_{cos}}\left(\left[\boldsymbol{h}_a^L, \boldsymbol{h}_b^L, \boldsymbol{h}_c^L, \boldsymbol{h}_d^L, \boldsymbol{z}_t\right]\right) \forall (a,b,c,d) \in \mathbb{T} \\
sin\psi_{abcd}^t &= \Pi_{tor_{sin}}\left(\left[\boldsymbol{h}_a^L, \boldsymbol{h}_b^L, \boldsymbol{h}_c^L, \boldsymbol{h}_d^L, \boldsymbol{z}_t\right]\right) \forall (a,b,c,d) \in \mathbb{T}
\end{aligned}
\tag{10}
$$

where $\Pi_*$ are two-layer MLPs with ELU activations and dropout that map from the concatenated node embeddings and latent state to the single scalar of interest. $\mathbb{B}$ denotes the set of all pairs of atoms defining a bond, $\mathbb{A}$ is the set of all triplets of atoms defining a bond angle, and $\mathbb{T}$ is the set of all quadruplets of atoms defining a torsion angle. Note that the decoder outputs a prediction for the bond angles directly, while for the torsion angles, $sin$ and $cos$ are predicted. This design choice is grounded on the fact that the models' parameters could not be optimized to decode the full space of torsion angles when predicting them directly.

## D  TRAINING AND INFERENCE

We define the reconstruction loss in terms of internal coordinates by:

$$
\begin{aligned}
\mathcal{L}_{rec}(\mathcal{G}_i, \widehat{\mathcal{G}}_i) = {}&\xi_b \frac{1}{|\mathbb{B}|} \sum_{(a,b)\in\mathbb{B}} ||d_{ab} - \hat{d}_{ab}|| \\
&+ \xi_a \frac{1}{|\mathbb{A}|} \sum_{(a,b,c)\in\mathbb{A}} cos(\phi_{abc} - \hat{\phi}_{abc}) \\
&+ \xi_t \frac{1}{2|\mathbb{T}|} \sum_{(a,b,c,d)\in\mathbb{T}} \left(cos(\psi_{abcd}) - cos\hat{\psi}_{abcd}\right)^2 + \left(sin(\psi_{abcd}) - sin\hat{\psi}_{abcd}\right)^2
\end{aligned}
\tag{11}
$$

where $\xi_b, \xi_a, \xi_t$ are hyperparameters, $\mathbb{B}$ denotes the set of all pairs of atoms defining a bond, $\mathbb{A}$ is the set of all triplets of atoms defining a bond angle, and $\mathbb{T}$ is the set of all quadruplets of atoms defining a torsion angle. Note that as described in subsection C.2, the model predicts the bond angles directly, whereas, for the torsion angles, it predicts $sin(\psi)$ and $cos(\psi)$.

To infer $\sigma^2$, i.e. the amount of noise added during inference, we found that the required noise level strongly correlates with the variance of the (normalized) torsion angles in the warmup trajectory. We identified a relationship of

$$
\sigma^2 = \frac{1}{|\mathbb{T}|} \sum_{i=1}^{|\mathbb{T}|} Var(\psi_i)
\tag{12}
$$

to reliably give a good estimate of the noise level with $|\mathbb{T}|$ being the number of torsion for the respective molecule. While this relationship holds across molecules, we used a noise level of $\sigma_i^2 = 6 * Var(\psi_i)$ for the alanine dipeptide model where the factor of six was inferred from the norm of the latent space.

## E  DATASET DETAILS

All datasets were created by performing MD simulations using the *openmm* library (Eastman et al., 2017).

The simulation was performed with the parameters shown in Table 2 and Figure 14 shows the free energy surface based on the two backbone dihedral angles $(\phi, \psi)$ of alanine dipeptide in implicit solvation. Given the distribution of $(\phi, \psi)$, the free energy surface can be computed by:

$$
FES_i = -k_B T \, ln\left[p(\phi_i, \psi_i)\right]
\tag{13}
$$

where $k_B$ is the Boltzmann constant and $T$ is the temperature of the system. We can observe five energetically favorable metastable states $\{P_{II}, C_7^{ax}, C_5, \alpha_R, \alpha_L\}$ which we also refer to as modes of the Boltzmann distribution. Note that the metastable states $\{C_7^{ax}, \alpha_L\}$ are visited rarely.

Table 2: Alanine dipeptide dataset properties.

| Property | Value |
|---|---|
| Simulation time | $100ns$ |
| Integrator | Langevin |
| Integrator time step | $1fs$ |
| Forcefield | AMBER ff96 |
| Solvation | OBC GBSA implicit |
| Frame Spacing | $100fs$ |
| Temperature | $300K$ |

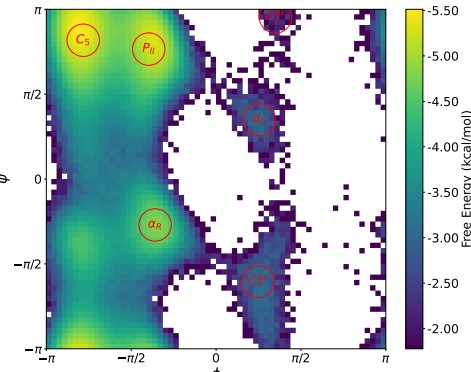

Figure 14: Ramachandran plot of the two backbone dihedral angles of the alanine dipeptide dataset with parameters from Table 2 and metastable states $\{P_{II}, C_7^{ax}, C_5, \alpha_R, \alpha_L\}$ as defined by Vlachas et al. (2022).

The dipeptide dataset was created with the simulation parameters given in Table 3.

Table 3: Dipeptide dataset properties.

| Property | Value |
|---|---|
| # Peptides | 216 |
| Simulation time (each) | $12ns$ |
| Integrator | Langevin |
| Integrator time step | $1fs$ |
| Forcefield | AMBER 14-all |
| Solvation | implicit GBn |
| Frame Spacing | $120fs$ |
| Temperature | $300K$ |

## F  IMPLEMENTATION DETAILS

All experiments were implemented in *PyTorch* (Paszke et al., 2019) using the extension for deep learning on graphs *Pytorch Geometric* (Fey & Lenssen, 2019). Furthermore, the *scipy* library (Virtanen et al., 2020) is extensively used throughout our implementation and we utilized the *stateinterpreter* package (Novelli et al., 2022) to automatically identify metastable states.

The experiments were run on two different machines. All training was run on a machine with two

AMD EPYC 7513 CPU @ 2.60GHz with 32/64 cores each, 504GB of RAM, and eight NVIDIA RTX A6000 GPUs with 48GB vRam of which only a single one was used at a time. All inference experiments were performed on a machine with two Intel(R) Xeon(R) Gold 6230 CPU @ 2.10GHz with 20/40 cores each, 504GB of RAM, and eight NVIDIA Tesla V100 GPUs with 32GB vRam where again only a single GPU was used at a time.

## G  ADDITIONAL MODEL VARIATIONS

**Dynamical Propagator** We found the LSTM architecture to consistently achieve the best simulation metrics outperforming the following architectures: Gated Recurrent Unit (GRU) (Cho et al., 2014); MLP; Mixture Density Network (Bishop, 1994); Transformer for time series forecasting (Wu et al., 2020). Besides the different architectures, we evaluated if conditioning the dynamical model onto the molecule it currently works with improves the generalization capabilities of our model. To do so, we employed another GNN that computes a fixed-size embedding based on the two-dimensional molecular graph, essentially constructing a learned representation of a certain molecule. This representation was then appended to the latent space to facilitate the prediction of correct dynamics for the propagator. However, we did not encounter any benefits of using this approach in terms of the quality of predicted trajectories for varying molecules.

**Training Schemes** Besides the training scheme described in subsection 3.2, we explored various methods of improving the robustness of the dynamical model mainly inspired by the approaches of Brandstetter et al. (2022). The model always gets correct latent states as input at training time whereas at inference time the propagator gets its own previous prediction as input which introduces a distribution shift between training and inference time. To mitigate this error, Brandstetter et al. (2022) suggest the "pushforward trick" which means to instead of using the correct latent state as input, the previous prediction of the dynamical model is used with a certain probability. Additionally, we tested whether infusing noise at different stages of our pipeline (in cartesian space; in internal coordinate space; in the latent space) improves the test performance of our dynamical model. While the above two approaches did not improve the simulation results, we found the approach of unrolling the LSTM for multiples of its sequence length and cutting the gradients between the steps as described in subsection 3.2 to be absolutely crucial for the model to learn correct long-term dynamics.

**Pretraining the autoencoder** In contrast to the results of Sidky et al. (2020), we found that pretraining the autoencoder did not improve simulation results but in fact significantly constrained the latent space such that dynamical properties could not be modeled precisely anymore.

## H  HYPERPARAMETERS

For all training, we use the *Adam*[9] optimizer and the *ReduceLROnPlateau*[10] learning rate scheduler with reduction parameter 0.7 and patience 5 epochs. We define an epoch to consist of 12 batches of trajectories with length $T$ for alanine dipeptide and 16 batches for the peptide models and train each model for 100 epochs, as we found all training metrics to have fully converged after that time.

Training the smaller model on alanine dipeptide took $14.6$ hours with a memory consumption of $8.9GB$. During inference, the memory consumption was $6B$, which is mainly caused by the batched decoding of structures where we used batches of size $1e5$ and which could be adapted to other hardware limitations. For the dipeptide models, training took approximately three days with a memory consumption of $43GB$. For decoding, we used a batch size of $1e4$, which led to $14GB$ of used GPU memory.

---

[9]https://pytorch.org/docs/stable/generated/torch.optim.Adam.html
[10]https://pytorch.org/docs/stable/generated/torch.optim.lr_scheduler.ReduceLROnPlateau.html

## H.1 ALANINE DIPEPTIDE HYPERPARAMETERS

The parameters were tuned in the order in which they appear in the table from top to bottom. The final parameters are marked in **bold**.
We found the batch size to have a significant impact on the performance of our model, as batches larger than 8 independent trajectories prevented the models to produce reasonable inference results. While we do not have concrete evidence, we suspect this to be the case because batches larger than 8 contain too diverse trajectories, essentially impeding the computation of meaningful gradients.

Table 4: Search space for the general hyperparameters, spanning across encoder, decoder and propagator.

| Parameter | Search Space |
|---|---|
| latent embedding dimension | [5, 10, 32, 64, 75, 100, 128, **256**, 512] |
| data normalization | [**min-max**, z-score] |
| batch size | [2, 4, **8**, 16, 32, 64] |
| starting learning rate | [1e-3, 5e-4, **1e-4**, 1e-5, 1e-6] |
| $c$ | [1, 2, 5, 10, 25, 50, 100, **120**, 150, 200] |
| $\delta_{rec}, \delta_{lat}, \delta_{e2e}, \xi_b, \xi_a, \xi_t$ | [0.33, **1** , 2] (independently altered) |

Table 5: Search space for the hyperparameters of the encoder network.

| Parameter | Search Space |
|---|---|
| # layers | [2, 3, 4, **5**, 6, 7, 8, 10] |
| # final MLP layers | [**1**, 2, 3, 4] |
| # attention heads | [2, 4, **8**, 16] |
| node embedding size | [5, **10**, 15, 25] |
| edge embedding size | [**2**, 4, 8, 12] |
| # readout function | [**Set2Set**, Sum, Mean] |
| dropout | [**0**, 0.1, 0.15, 0.2] |

Table 6: Search space for the hyperparameters of the decoder network.

| Parameter | Search Space |
|---|---|
| # MP layers | [1, 2, 3, 4, **5**, 6, 7, 8, 10] |
| k-hop edge concatenation | [**2**, 3, 4] |
| # attention heads | [2, **4**, 8, 16] |
| input node embedding size | [5, **10**, 15, 25] |
| output node embedding size | [10, 15, **25**, 50, 100] |
| # final MLP layers | [1, 2, **3**, 4] |
| dropout MP layers | [**0**, 0.1, 0.15] |
| dropout MLP layers | [0, **0.1**, 0.15] |

## H.2 DIPEPTIDE HYPERPARAMETERS

For the training of the peptide models, we identified a batch size of 64 to yield the best results.

Table 7: Search space for the hyperparameters of the LSTM propagator.

| Parameter | Search Space |
|---|---|
| $k$ (sequence length) | [1, 3, **5**, 10, 25, 50, 100, 250] |
| # LSTM layers | [1, 2, **3**, 4, 5, 6] |
| # MLP layers | [1, **2**, 3] |
| LSTM dropout | [0, **0.1**, 0.2] |
| $\beta$ | 0.15 |

Table 8: Search space for the hyperparameters of the dipeptides model. All hyperparameters that are not explicitly listed are the same as for the alanine dipeptide model.

| Parameter | Search Space |
|---|---|
| latent embedding dimension | [128, 256, 512, **1024**, 2048] |
| # num encoder layers | [4, 5, 6, **8**, 10] |
| # num decoder layers | [4, 5, 6, **8**, 10] |
| # LSTM layers | [ 4, **5**, 6, 8] |
| $c$ | [1, 2, 5, 10, 25, 50, **100**, 120, 150, 200] |
| decoder output node embedding size | [10, 15, 25, **50**, 100] |
| $\beta$ | 0.9 |

