# OpenReview forum: "Latent Space Simulator for Unveiling Molecular Free Energy Landscapes and Predicting Transition Dynamics"
_ICLR.cc/2024/Conference — ICLR 2024 Conference Withdrawn Submission_

### Official Review · Reviewer_aLv7 · 2023-10-23

**Soundness:** 2 fair
**Presentation:** 2 fair
**Contribution:** 3 good
**Rating:** 3
**Confidence:** 5

**Summary:**

In this work the authors propose a method for determining Free Energy Surfaces and deriving dependent quantities such as transition rates. Determining these properties in a computationally efficient manner is possibly an important contribution to drug and material design. The proposed method achieves a speedup of 20 over standard MD approaches by reducing the full atomistic state to a latent space and propagating the dynamics there based on a learned LSTM instead of the original bias potential. The authors evaluate their method on alanine dipeptide and a number of other dipeptides to show the generalisability of the method.

**Strengths:**

The presented paper looks at an important problem for drug and material design and is in general written in a pleasant style. The authors clearly and concisely describe their method.

**Originality:** The method shares a number of similarities with Coarse Graining and other methods that aim to reduce the dimensionality of a system during a dynamical simulation. As such, the method is not hugely original. However, the introduction of the SE(3)-invariance in this mapping does add some novelty.

**Quality:** The paper is missing some key citations and in general has some issues in its experimental evaluation. Additionally, the lack of details regarding the MD simulation data also brings down the quality of the paper.

**Clarity:** The paper is well written and the method is clearly introduced.

**Significance:** The paper addresses and important problem and thus has the potential to be significant. However, the results need to be improved to make sure that the presented claims are substantiated.

**Weaknesses:**

The paper insufficiently formalises some important concepts and this results in some misunderstandings and/or unclarity. As a result of this it is hard to judge the full value of the presented results. I've made more detailed comments about where these issues occur in the comments and questions section below.

**Questions:**

Please find my detailed list of comments, ordered by section, below.

Where needed, I indicated that something might be a breaking issue. These issue will have to be either clarified or rectified before I will consider increasing my vote to accept.

**Introduction:**
1. When discussing the possible applications of FES estimation, it would be in my opinion more appropriate to cite survey articles instead of 2 single application papers.
2. In the second paragraph the authors introduce the concept of the FES, stable states, transition pathways and free energy differences but insufficiently introduce how these concepts are related. Additionally, the authors introduce the transition rates as a separate entity entirely unrelated to the FES. However, the transition rate is in itself a function of the FES and should as such no be treated differently. Concretely; It would be beneficial to see a more formalised section introducing the FES as a function of the partition functions and directly relating the introduce properties to this definition.
3. While the authors provide a broader picture in the related work section, the discussion of the usual approach for estimating the FES in paragraph 3 is insufficient. At this point in the paper there should already be a reference to topics such as Free-Energy pertubation methods, metadynamics and Transition Path Sampling.
4. The criteria for a FES estimation method introduced in paragraph 3 are in my opinion not concrete enough and therefore lack actionability. Concretely; For point 1, how does this relate to FES that do directly model the distribution and not based on simulation? For point 2, the authors have not yet introduced the concept of reaction coordinates and should do so together with the FES introduction. For point 3, a more formal definition of "unphysical states" is needed.
5. Multiple works have come out lately that generalise beyond single systems. Eg. [1] and the later mentioned [Zheng et al., 2023]. The authors should mention these, possibly as an indicator for why extending LED in this way is important.
6. Regarding the sentence; "Other approaches [....], do not have a notion of time, i.e., no dynamical properties such as the transition rates can be extracted." This is mentioned a couple of time throughout the paper and signals a clear misunderstanding. Reaction Rates/Transition Rates/MFPT are all functions of the FES and therefore, once the FES has been established, do not depend on a dynamical system to determine. Concretely; the authors should clarify if I'm misunderstanding their point or rectify the manuscript. This is a breaking issue.
7. When discussing machine learning approaches for FES estimation, the authors fail to mention a number of recent papers that are similar to their work. Ie. they speed up simulation using machine learning either by introducing a larger time-step or introducing a bias force. Eg: [1] and [2]

**Related work:**
1. Regarding Course Graining the authors claim that "Their accuracy is usually not sufficient for recovering the FES of flexible molecules.". This claim should be substantiated with either experiments of references. As of now, it is also not clear if LaMoDY solves these issues. Similarly, the authors claim that "LaMoDy generalises across systems" as an improvement over CG. However, Coarse Graining is in general a method that generalises to other systems for which the same mapping can be used very well. Concretely; the authors should substantiate their criticism of CG using citations or experiments. Preferably, CG would be used as a benchmark for the experimental evaluation of the proposed method to show the mentioned issues with CG.
2. The authors mention "they typically require determining collective variables (CVs)" as one of the challenges of enhanced sampling methods. However, for projecting the FES the method presented relies on dihedral angles as CV as discussed in the last sentence of page 5. Concretely; the authors should put more emphasis on LaMoDy no needing to specify CVs BEFORE the sampling procedure. This is in itself a valuable contribution, but the distinction should be made more clear.
3. Related, the paper fails to mention other enhanced sampling methods that do not require CV such as Transition Path Sampling [3, 4], Nudge Elastic Band [5], and more recent ML based methods [6, 7, 8].
4. Lastly, regarding enhanced sampling methods, similar to the breaking issue I mentioned in the introduction, despite not having a "explicit notion of time", enhanced sampling methods are often used for determining kinetic properties such as reaction rates [9, 10, 11].

**Method**
1. From my understanding the following is my best high-level summary of the method proposed by the authors; "MaLoDY learns to reduce the full atomic system to a reduced space that is sufficient to model most degrees of freedom important for the FES (as such, this is very similar to Coarse Graining and other CV learning methods such as TICA). Using the reduced representation, MaLoDY learns a recurrent model to predict for each state the most likely next state given the predetermined timestep. While the timestep is equally small as the timestep needed for the MD simulation itself, predicting the next state using the LSTM on the reduced latent space is quicker than performing the MD update using the potential." Concretely; Is this characterisation of the method correct?
2. Why is the representation used for the systems different between the encoder and decoder. As I understand, the encoder currently uses the bonds as nodes for the graph while the decoder uses the atoms as nodes?
3. The authors state: "As suggested by Vlachas et al. (2022), sequences of MD states are not necessarily Markovian since complex systems can exhibit long-term correlation in their behaviour, [...]". It is important to note here that this is only true when the simulation does not model all degrees of freedom. When modelling a system with implicit solvent and Langevin Dynamics, which I assume has been done in this work, the system is fully Markovian.
4. In section 3.2 the authors introduce $\mathcal{E}$ for denoting the encoder and $\mathcal{D}$ for the decoding. These should be introduced more explicitly.
5. There is a lot of information missing about the training data used and the general setup of the MD simulation that was run to obtain it. For example, it is not clear if the training data was obtained using some form of stochastic thermostate? Concretely; This is a breaking issue. To be able to accept the paper information is needed about the framework, the forcefield and the integrator used for the MD simulations, as well as basic information such as discretisation and temperature.
6. When introducing the inference scheme, the authors mention: "Additionally, we infuse artificial noise to the latent space before feeding them into the propagator. We found this to be crucial because otherwise the dynamical model was prone to become stuck at a local energy minimum." This changes the thermodynamical properties of the system and should therefore be discussed in more depths. By introducing this we will make arbitrary changes between meta-stable states that changes the computed free energy difference between them. Concretely; This is a breaking point and should be addressed by the authors.

**Evaluation Protocol for FES recovery**
1. At the end of page 5 the authors address that the dihedral angles will be used to project the FES and obtain the Ramachandran plot. This is a crucial consideration and comes with a number of trade-offs. Most notably, it reduces the introduced metric to only be usable for dipeptides. This is acceptable for the work in question by should be addressed more clearly in the paper.
2. In figure 5 there are additional meta-stable states marked along the the bottom of the figure. These are incorrect and the results of the ramachandran plot being cyclical. This is a potentially breaking issue as it indicates that the metric used for extracting the meta-stable states is not correct and thus the results invalid. This is a potentially breaking issue. The authors will need to comment on this.

**Experimental Results**
1. The authors state: "Starting from an intialization sequence of five state, we simulate a trajectory of length 100ns without re-initialization", it is not fully clear what the authors mean by this. Does this refer to the number of timesteps the LSTM has to warm up using ground-truth MD data? Related, it would be important to specify what the starting location is for the sampling procedure.
2. The paper states: "[..] our model is able to capture all metastable states [..]". Given that we are looking at FES, it is also important to consider the accuracy of the FES at each state. While the authors mention that the "our model slightly overestimates the density of \alpha_R" they should also discuss the undersampling of other states.
3. The paper states "[..] no unphysical states are visited throughout the entire simulation. " As also mentioned in my discussion of the introduction, it is unclear what is meant with "unphysical states". My initial guess would be that unphysical states are states with unrealistically high potential energy, but this is not something we can derive from figure 7.
4. Based on figure 8, the paper states "The transition probabilities clearly shows that the dynamical properties that can be inferred from the model predictions closely match the true dynamics.". However, based on my own observations of the figure, this does not seem to be the case. For example, the true MD data shows a non-zero probability of moving from C_7 to C_5, while this is zero for the predicted data. This is a breaking issue.
5. For the 16 out of 216 possible peptides, it should be made more clear how different they are to the 200 training samples.
6. While the authors claim that "the learned simulator is more 'conservative' and avoids predicting unphysical modes rather than exploring the full state space which is desirable" I would argue that this in undesirable behaviour as it reflects that the trained simulator is not able to successfully bridge high free energy barriers. To truthfully evaluate the performance increase of the proposed method over standard MD simulation the paper should compare the presented results against the ground-truth simulation data of these dipeptides. Concretely; without a comparison agains the MD data the results are inconclusive. This is a breaking issues.
7. For figure 10 it states that "peptides that only contain one mode are excluded, as the MFPT error would be 0 in this case." Figure 10 shows that this is the case for 4 peptides (12 out of 16 test samples are shown). As such, for all these 4 peptides the recall of meta-stable states using MaLoDY should either be 0 or 1 as the meta-stable state is either found or not. However, in the presented results in figure 9, none of the peptides have a recall of 0 or 1. Concretely; The authors should clarify if I'm misunderstanding or if there is an issue with the evaluation. This is a breaking issue.
8. Regarding the final ablation study. Based on the final sentence presenting the authors view on how the differences occurred, it is unclear to me if these results have any value. As it stands, the results can entirely be contributed to the length of the unroll as with shorter unrolls it is unlikely that the method will observe changes between meta-stable changes within a single unroll.

**Discussion**
1. The discussion states "we introduce LaMoDY, a learned simulator operating in a latent space to efficiently recover free energy surfaces". However, the evaluation has fully focussed on determining meta-stable states and specifically did not use KL-divergence based methods to evaluate the full FES. As such, this claim should be altered in my opinion.

**References**

[1] https://arxiv.org/abs/2302.01170

[2] https://pubs.acs.org/doi/full/10.1021/acs.jctc.3c00702

[3] http://aip.scitation.org/doi/10.1063/1.475562

[4] https://www.annualreviews.org/doi/abs/10.1146/annurev.physchem.53.082301.113146

[5] https://pubs.aip.org/aip/jcp/article/113/22/9901/185050/A-climbing-image-nudged-elastic-band-method-for

[6] https://pubs.aip.org/aip/jcp/article/155/13/134105/353217/Reinforcement-learning-of-rare-diffusive-dynamics

[7] https://arxiv.org/abs/2207.02149

[8] https://arxiv.org/abs/2301.03480

[9] https://www.researchgate.net/publication/224877216_Transition_Path_Sampling_and_the_Calculation_of_Rate_Constants

[10] https://onlinelibrary.wiley.com/doi/10.1002/9781118889886.ch1

---

### Official Review · Reviewer_4Vhr · 2023-10-27

**Soundness:** 2 fair
**Presentation:** 3 good
**Contribution:** 2 fair
**Rating:** 3
**Confidence:** 4

**Summary:**

The paper introduces a LaMoDy as a novel strategy to accelerate molecular dynamics simulation based on a SE(3) invariant graph encoder, an LSTM latent propagator, and a graph decoder architecture. An experimental evaluation assesses the effectiveness of LaMoDy in identifying meta-stable states and accuracy of the transition dynamics for dipeptides observed during training and novel structures that were not part of the training set.
The authors also demonstrate that LaMoDy can accelerate molecular simulations by a factor 20x in the reported experiments.

**Strengths:**

1) The paper proposes and demonstrates the effectiveness of a model that can have relevant implications in accelerating Molecular Dynamics.

2) To the best of my knowledge, this paper first assesses and demonstrates the capability of deep-learning-based architectures to unseen molecular structures.

3) The paper is generally well-structured and accessible.

**Weaknesses:**

## Main Concerns
1) **Novelty** The paper does not introduce a fundamentally novel methodology since the latent simulation strategy is similar to previous work [1,2]. The main differences with [1] are not sufficiently addressed in the main text.


2) **Experimental evaluation**
    1) **Hyper-parameter importance** The proposed LaMoDy model introduces a number of hyper-parameters controlling the importance of several loss components $\delta_{e2e}$, $\delta_{lat}$, $\delta_{rec}$, the stochasticity in the latent transitions $\beta$ and $\sigma$, and the weights for the reconstruction terms $\xi_b$, $\xi_a$, and $\xi_t$.
Appendix H reports the hyper-parameters used for the reported experiments but lacks a description of the hyper-parameter selection procedure. More importantly, parameters such as $\beta$ and $\sigma$ seem to be essential in determining the unfolded dynamics, but no ablation study or further intuition is provided.

    2) **Evaluation metrics** The paper claims that one of the contributions is "defining a systematic evaluation scheme". Nevertheless, the strategy of using the two torsion angles $\phi$ and $\psi$ to define the FES seems applicable only to dipeptides, since larger proteins or different molecules would not be fully characterized by the proposed description. Furthermore, this procedure can not account for errors in reconstructed bond angles and bond length.


    3) **Lack of baselines** The paper includes qualitative visualizations and quantitative measurements for LaMoDy, but no other model or baseline is included in the comparison. Even if other models in the literature can not be straightforwardly applied to unseen molecules, the paper can still assess the effectiveness of the proposed changes in the Alanine Dipeptide experiments.

    4) **Lack of experimental details and results** Little details regarding the peptide dataset used in section 5.2 are provided. The paper does not include any qualitative visualization of FES other than Alanine Dipeptide nor shows reconstruction (or FES) for any of the molecules used in section 5.2. The results reported in Figure 9 and Figure 10 are quite limited to support the claim that LaMoDy can effectively generalize to unseen molecules, especially considering that no measure of standard deviation is provided.


My main concerns regarding the submission revolve around the novelty and experimental evaluation. I believe that the paper sensibly addresses a relevant problem, but it should either introduce a novel well-motivated aspect or provide a stronger experimental analysis to argue for acceptance.

## Minor Issues
1) The notation in section 3.1 is not exhaustively described. The paper defines the bond lengths $\mathbb{B}$, bond angles $\mathbb{A}$, and torsion angles $\mathbb{T}$, but the meaning of $\mathcal{V}$, $\mathcal{B}$, $\mathcal{X}$, and $\mathcal{C}$ is not entirely clear from the context. The **Encoder** paragraph is missing the reference to Appendix A.1.

## References
[1] Vlachas, Pantelis R., et al. "Accelerated simulations of molecular systems through learning of effective dynamics." Journal of Chemical Theory and Computation 18.1 (2021): 538-549.

[2] Sidky, Hythem, Wei Chen, and Andrew L. Ferguson. "Molecular latent space simulators." Chemical Science 11.35 (2020): 9459-9467.

**Questions:**

1) What are the crucial differences between LaMoDy and the work in [1,2]? Do they translate into fundamental differences in performance according to the proposed metrics?

2) What is the effect of the various hyper-parameters for LaMoDy? In particular, $\beta$ and $\sigma$ seem to determine the trajectory of the unfolded simulation. How does changing them affect the resulting simulations qualitatively and in terms of the proposed metrics?

3) How does LaMoDy perform when compared to other models in literature in the setting in which train and test sets use trajectories from the same molecule?

4) In which situations does LaMoDy fail to accurately model the dynamics? Do failure cases correlate with some structural properties of the simulated peptides?

5) The proposed evaluation metrics seem to solely focus on the reconstructed torsion angles. What role does the reconstruction error for bond lengths and angles play?

6) How does the performance of LaMoDy change across different train/test splits and initialization?

---

### Official Review · Reviewer_HMfm · 2023-10-27

**Soundness:** 2 fair
**Presentation:** 2 fair
**Contribution:** 2 fair
**Rating:** 5
**Confidence:** 5

**Summary:**

The paper presents LAMODY, a latent-space molecular dynamics (MD) simulator for estimating Free Energy Surfaces (FES) and metastable transition rates in molecular systems. The proposed approach leverages an achirality-aware SE(3)-invariant encoder-decoder architecture to generate a latent space and a recurrent neural network to run the time-wise dynamics.

**Strengths:**

* The motivation for developing a more efficient MD simulator is well-established, and the problem of intractability in large systems or databases is clearly presented.
* The paper is well-written and structured, providing sufficient background information and a clear description of the proposed method.

**Weaknesses:**

The related issues will be elaborated in the "questions" section.

**Questions:**

* The paper primarily focuses on the application of existing techniques to a chemical problem domain, rather than proposing novel algorithmic advancements. This may be suitable for a journal rather than a machine learning conference. Or can the authors provide more insights on the algorithmic innovations that may be relevant to the machine learning community in this field?
* Can the authors provide a more detailed comparison with existing methods, particularly discussing the advantages and limitations of the proposed LAMODY model compared to other state-of-the-art approaches? Furthermore, I also find a related paper[1].
* Will LAMODY model meet unphysical conformations during the simulation? How will it be handled?

[1] Xie T, France-Lanord A, Wang Y, et al. Graph dynamical networks for unsupervised learning of atomic scale dynamics in materials[J]. Nature communications, 2019, 10(1): 2667.

---

### Official Review · Reviewer_1u1D · 2023-10-30

**Soundness:** 2 fair
**Presentation:** 3 good
**Contribution:** 2 fair
**Rating:** 3
**Confidence:** 4

**Summary:**

The authors propose LaMoDy, a latent-space MD simulator for molecular conformation exploration. They utilize SE(3)-invariant GNN encoder/decoder and LSTM as the latent space propagator to update molecular conformation representations in latent space, possibly corresponding to important collective variables. The major contributions include (1) a stable latent-space propagator with long-term stability; (2) the model is generalizable to unseen molecules (dipeptide systems); (3) proposed a new metric (MSPR) for FES recovery tasks.

In general, the manuscript is easy to follow, where the main ideas are clearly presented. However, I have some major concerns regarding the alleged generalizability and the novelty of the proposed metric.

**Strengths:**

- Effectively exploring molecular FES is a fundamental yet challenging task. Deep learning models have the potential to capture significant collective variables and dynamic modes of molecules through training. LaMoDy presents a clear and reasonable workflow towards solving this grand challenge.
- Developing well-defined/carefully-designed benchmark datasets and evaluation metrics is crucial for the research community to fairly compare model effectiveness. This manuscript brings about this issue and proposes a new metric.

**Weaknesses:**

- Only used dipeptide as benchmark data, thus cannot guarantee generalizability to other molecular systems. See questions.
- Lack of fair comparison with existing learning-based models. See questions.
- The proposed metric does not seem to be novel, but a composition of existing methods. See questions.

**Questions:**

- Can you provide comparison between LaMoDy and other existing models, e.g., Timewarp, on the same benchmark datasets? That would make the model performance/advantage more convincing.
- The model has only been evaluated on dipeptides, where different molecules share similar collective variables (dihedral angles). More evidence should be provided on different molecules to demonstrate model generalizability.
- Fig. 9, the training set metrics seem to be quite poor. The model is not doing very well even during training. How do you explain this behavior?
- The proposed MSPR is based on Novelli et al., after which precision and recall metrics are calculated. I find very limited novelty here. In addition, MSPR is not even clearly defined in the main text.
- How does model performance change on these dipeptides as a function of time step (i.e., compared with 100 fs results)?
- During inference, artificial noise is added to the latent representation (Eq. 3). Such noise is not used during training. I suppose the model should be able to learn the "drift" and "diffusion" terms from training, so why is it necessary to add this extra step only at inference time?
- Page 2, end of Introduction, "that allow that reflect", fix typo.